# Litterbox—A gnotobiotic Zeolite-Clay System to Investigate Arabidopsis–Microbe Interactions

**DOI:** 10.3390/microorganisms8040464

**Published:** 2020-03-25

**Authors:** Moritz Miebach, Rudolf O. Schlechter, John Clemens, Paula E. Jameson, Mitja N.P. Remus-Emsermann

**Affiliations:** 1School of Biological Sciences, University of Canterbury, 20 Kirkwood Avenue, Christchurch 8053, New Zealand; moritz.miebach@pg.canterbury.ac.nz (M.M.); rudolf.schlechter@pg.canterbury.ac.nz (R.O.S.); john.clemens@canterbury.ac.nz (J.C.); paula.jameson@canterbury.ac.nz (P.E.J.); 2Biomolecular Interaction Centre, University of Canterbury, 20 Kirkwood Avenue, Christchurch 8053, New Zealand

**Keywords:** gnotobiota, microbe–microbe interactions, phyllosphere, plant immunity, plant microbiota, plant–microbe interactions, rhizosphere, single-cell, synthetic community

## Abstract

Plants are colonised by millions of microorganisms representing thousands of species with varying effects on plant growth and health. The microbial communities found on plants are compositionally consistent and their overall positive effect on the plant is well known. However, the effects of individual microbiota members on plant hosts and vice versa, as well as the underlying mechanisms, remain largely unknown. Here, we describe “Litterbox”, a highly controlled system to investigate plant–microbe interactions. Plants were grown gnotobiotically, otherwise sterile, on zeolite-clay, a soil replacement that retains enough moisture to avoid subsequent watering. Litterbox-grown plants resemble greenhouse-grown plants more closely than agar-grown plants and exhibit lower leaf epiphyte densities (10^6^ cfu/g), reflecting natural conditions. A polydimethylsiloxane (PDMS) sheet was used to cover the zeolite, significantly lowering the bacterial load in the zeolite and rhizosphere. This reduced the likelihood of potential systemic responses in leaves induced by microbial rhizosphere colonisation. We present results of example experiments studying the transcriptional responses of leaves to defined microbiota members and the spatial distribution of bacteria on leaves. We anticipate that this versatile and affordable plant growth system will promote microbiota research and help in elucidating plant-microbe interactions and their underlying mechanisms.

## 1. Introduction

Plants offer three different habitats to microbes: the endosphere, the rhizosphere, and the phyllosphere. The endosphere encompasses the habitat formed by the internal tissues of plants, whereas the rhizosphere and phyllosphere encompass the surfaces of belowground and aboveground plant organs, respectively. Plants are hosts to remarkably diverse and complex, yet structured, microbial communities, collectively referred to as the plant microbiota [1,2,3]. Owing to the diversity and complexity of the microbiota, it is not surprising that the traditional view of host–microbe interactions has tended to focus on important but discrete aspects of the whole, such as plant pathogens, nitrogen-fixing rhizobacteria, and phosphate-mobilizing mycorrhizal fungi. This view has recently shifted to a holistic one considering the plant and its associated microbiota as a metaorganism or holobiont [4,5,6]. It is widely recognised that members of the microbiota assist in nutrient uptake, promote growth, and protect against biotic and abiotic stresses [7,8,9,10,11,12,13,14]. To harness these positive impacts of plant-associated microbiota, the use of synthetic microbiota has been proposed [15,16,17]. The prospect of using synthetic microbial communities to promote sustainable agriculture is leading to a growing appreciation of plant microbiota research.

Generally, plant microbiota research aims to understand plant–microbe and microbe–microbe interactions ranging from the individual microorganism to the microbial community resolution. It also seeks to understand the underlying molecular mechanisms of these interactions, and their contribution to microbial community assemblage. Thus far, most of our understanding of microbial community assemblage and function is derived from observations at the whole community level (including 16S rDNA and ITS sequencing, meta-genomics, -transcriptomics, and -proteomics) [18,19,20,21,22]. Such observations showed that microbial community assemblage is governed by many factors, including abiotic impacts, physicochemical properties of the leaf, plant–microbe, and microbe–microbe interactions [2]. However, owing to the multitude of factors affecting community assemblage in parallel, their effects are often convoluted. Therefore, reductionist approaches are needed to disentangle the complexity and to allow a shift from correlation to causation [23]. Bai et al. (2015) have built an extensive indigenous bacterial culture collection covering phyllosphere and rhizosphere inhabitants of *Arabidopsis thaliana*, enabling the research community to study the functions of individual microbiota members in synthetic communities [24]. While synthetic communities might not mimic nature perfectly, they allow for testing of causality either by bottom-up (adding microorganisms to low complexity synthetic communities) or drop-out (excluding single strains from rather complex synthetic communities) experiments [23,25,26]. Furthermore, to fully understand plant–microbe and microbe–microbe interactions in the heterogeneous phyllosphere, the interactions need to be studied at micrometre resolution, the scale relevant to microbes [1].

A deeper understanding of the driving factors of microbial community structures will enable us to engineer synthetic microbial communities that are both plant protective and resilient. In addition, to engineer synthetic microbial communities that are superior to individual biocontrol strains, knowledge about the varying modes of plant protection by biocontrol strains and the compatibility of these biocontrol strains is essential. For example, it was shown that *Pseudomonas fluorescens* A506 can alleviate the biocontrol activity of *Pantoea vagans* C9-1, whereas each is individually effective as a biocontrol strain against *Erwinia amylovora*, the causal agent of fire blight [27]. *Pseudomonas fluorescens* A506 produces an extracellular protease that degrades the peptide antibiotics produced by *Pantoea vagans* C9-1 [28]. Therefore, to fully harness the power of the microbiome, our understanding of the varying modes of biocontrol activity needs to be extended by studying the underlying mechanisms of plant–microbe and microbe–microbe interactions. To do so, hypothesis-driven research needs to be conducted in a highly controlled, yet adjustable gnotobiotic, i.e., otherwise sterile, system.

Highly controlled and reproducible growth conditions are needed as abiotic factors can influence the plant’s response to its microbial colonisers [29,30]. Furthermore, pH strongly influences soil microbial communities [31,32]. Controlling abiotic factors alleviates confounding effects of these on microbial community structure. Controlled and reproducible growth conditions also allow for time-course experiments, which are important as plant responses to microbial colonisers are dynamic [33,34]. The order in which microbes are introduced, as well as genetic alterations of the host or the microbes, allows for a mechanistic understanding of plant–microbe and microbe–microbe interactions.

Arabidopsis can be grown gnotobiotically under tissue culture conditions on a variety of different substrates. Phytoagar is widely used, allows for reproducible growth, and is easy to handle. Phytoagar-based systems have also been successfully used for plant–microbe interaction studies [9,35]. However, aeration of the root system is sub-optimal and nutrients are non-uniformly delivered to the root surface, rendering the system highly artificial [36]. Hydroponics and aeroponics allow for a uniform distribution of nutrients and O_2_, but the lack of structure leads to abnormal root phenotypes [36]. This can be circumvented by the use of filter paper in a hydroponic system [37]. However, aeroponic and hydroponic-based systems are often labour intensive to set up. Furthermore, plants grown in the systems mentioned above are exposed to high to very-high humidity, which cannot be modulated over a wide range. The microclimate in which a plant grows severely affects leaf cuticle composition leading to differences in water permeability and leaf wettability and finally to shifts in microbial community structure [38,39,40]. Humidity has been described as one of the main drivers for bacterial colonisation on plants, likely affecting the bacterial load [41].

With the aim of minimising experimental artefacts and mimicking natural conditions, soil has been used in axenic plant growth systems. However, sterilisation of soil is difficult and is usually accompanied by structural and geochemical changes [42,43]. Recently, calcined clay has been used as an alternative to soil, rendering a similar texture to soil, but adjustable in its nutrient profile [44,45]. However, the desorptive properties of calcined clay can lead to toxic levels of labile Mn ions [46].

Here we introduce “Litterbox”, a gnotobiotic plant growth system that uses zeolite as a soil-like substrate. Zeolite is easy to sterilise, its porous structure ensures root aeration, it has an excellent water- and nutrient-holding capacity, it is known for its slow-release of nutrients, including Mn, thereby alleviating the risk of Mn toxicity [47,48], and it has previously been used in other plant growth systems, and even on space missions [49]. Additionally, by adding a sheet of polydimethylsiloxane (PDMS) as a physical barrier onto the surface of the zeolite, cross-inoculation of phyllosphere and rhizosphere resident microbes can be significantly reduced. Zeolite can be easily and affordably sourced from commercial “cat litter”.

## 2. Materials and Methods

### 2.1. Plant Growth

Seeds of *Arabidopsis thaliana* (Col-0) were surface-sterilised according to Lundberg et al. (2012) [50]. Depending on the experiment, seeds were either sown directly on zeolite (sourced from cat litter – Vitapet, Purrfit Clay Litter, Masterpet New Zealand, Lower Hutt, New Zealand), on ½ Murashige and Skoog medium (MS, including vitamins, Duchefa, Haarlem, Netherlands) 1% phytoagar (Duchefa, Haarlem, Netherlands) plates, pH 5.9, or on pipette tips (10 µL or 200 µL) filled with ½ MS 1% phytoagar, pH 5.9. Seeds were germinated in a CMP6010 growth cabinet (Conviron, Winnipeg, Canada) with 11 h of light (150–200 µmol m^−2^ s^−1^) and 85% relative humidity. Seedlings germinated on agar plates or agar-filled pipette tips were transferred seven days after sowing to autoclaved plant tissue culture boxes (Magenta vessel GA-7, Magenta LLC, Lockport, IL, USA), filled with either 90 g fine ground zeolite or 70 g coarse zeolite, and covered by 20 g finely ground zeolite, soaked with 60 ml of ½ MS including vitamins, pH 5.9. Fine ground zeolite was obtained using a jaw crusher (Boyd Crusher, Rocklabs, Scott Technology Ltd – Automation & Robotics, Dunedin, New Zealand). The zeolite was covered with a PDMS sheet (thickness 0.5–1.5 mm) to reduce inoculation of the growth substrate with bacteria. Lids of the plant tissue culture boxes contained four holes for gas exchange (9 mm diameter), which were covered by two pieces of micropore tape (3M) to ensure axenic conditions. Plants were grown in a growth room at 21 °C and 11 h of light (120–200 µmol m^−2^ s^−1^). Plant tissue culture boxes were put into transparent plastic containers (Sistema Plastics Limited, Auckland, New Zealand) and closed with cling wrap to maintain high relative humidity. After inoculation, the plant tissue culture boxes were kept in a CMP6010 growth cabinet with 11 h of light (150–200 µmol m^−2^ s^−1^) and 85% relative humidity. High batch-to-batch variation in readily available ions was reported in calcined clays leading to heavy metal toxicity [46]. We also observed plant phenotypes resembling heavy metal toxicity in one out of ten batches of cat litter. It is thus advisable to test the growth of a few plants when switching to a new batch of cat litter. 

### 2.2. Fabrication of PDMS Sheets

PDMS and catalyst (Sylgard 184, Dow Corning, Midland, Michigan, United States) were mixed at a 10:1 ratio and the solution was vigorously mixed using a plastic rod. To release trapped air bubbles, the mix was evacuated, and the vacuum was released several times until no more air bubbles appeared. Two A4 acetate sheets (Overhead Projector Transparency Film, OfficeMax, Office Depot, Inc., Boca Raton, FL, USA), were placed on a flat-surfaced sandwich press (GR8450B, Sunbeam Products, Boca Raton, FL, USA) with an area of approximately 720 cm^2^. The PDMS (44 mL) was poured on top and evenly distributed with a spatula. The top of the sandwich press was lowered to about 9 mm above the PDMS. The PDMS was cured at 60 °C for 1 h. The PDMS was then carefully pulled from the acetate sheets and cut to 6 × 6 cm squares. A hole punch was used to punch four holes into each PDMS sheet, to allow the growth of four plants per plant tissue culture box. 

### 2.3. Plant Inoculation

The two leaf colonising model strains *Sphingomonas melonis* FR1 and *Pseudomonas syringae* DC3000 (Table 1) were cultivated at 30 °C on R2A (HIMEDIA LABORATORIES, Mumbai, India) or King’s B (HIMEDIA LABORATORIES, Mumbai, India) media plates, respectively. Anticipating subsequent plant gene expression analysis, the bacterial strains were cultivated on minimal media [51] agar plates containing 0.1% pyruvate as a carbon source to prevent the contamination of plants with growth media-derived cytokinins that may impact plant responses [52]. Bacterial suspensions were prepared from bacterial colonies suspended in phosphate-buffered saline (PBS, 0.2 g L^−1^ NaCl, 1.44 g L^−1^ Na_2_HPO_4_ and 0.24 g L^−1^ KH_2_PO_4_) and washed twice via centrifugation at 4000× *g* for 5 min followed by discarding the supernatant and adding PBS. The bacterial suspensions were then diluted to a desired optical density (OD_600nm_ = 0.025). Next, 200 µL of bacterial solution was sprayed per plant tissue culture box using an airbrush spray gun (0.2 mm nozzle diameter, Pro Dual Action 3 #83406, Buyeasy Inc Store, AliExpress, Store No. 1680024, Shanghai, China). To obtain a homogeneous coverage, the distance between the airbrush spray gun and the plants was increased by stacking a bottomless plant tissue culture box on to the plant tissue culture box containing the plants being spray inoculated.

To co-inoculate cyan fluorescent protein, mTurquoise2, expressing *S. melonis* Fr1 (*S. melonis* Fr1::mTurquoise2) and red fluorescent protein, mScarlet-I, expressing *P. syringae* DC3000 (*P. syringae* DC3000::mScarlet-I) (Table 1) onto *A. thaliana* leaves, a bacterial suspension was prepared as described above to a final optical density (OD_600nm_) of 0.005 (1:1 ratio) before being airbrushed onto 47 days-old Arabidopsis. Fluorescently tagged strains were constructed as described previously [53].

### 2.4. Enumeration of Bacteria Recovered from Varying Environments

Bacterial colony forming units (cfu) in different environments (phyllosphere, rhizosphere, zeolite) were determined. To this end, aboveground plant parts were carefully detached from the belowground parts and placed individually in 1.5 mL tubes. To sample the rhizosphere, the zeolite containing the roots was poured out and the roots were cut and placed individually in 1.5 mL tubes. A spatula was then used to collect between 100 and 300 mg of zeolite, which was placed individually in 1.5 mL tubes. After determining the plant or zeolite sample weight, 1 mL PBS with 0.02% Silwet L-77 (Helena chemical company) was added to each tube. The bacteria were then dislodged from the sample via shaking twice for 5 min at 2.6 m s^−1^ in a tissue lyser (Omni Bead Ruptor 24), followed by sonication in a water bath for 5 min. Three microlitre spots of tenfold dilutions of each sample were spotted onto R2A plates. Cfu were counted after incubation of the plates at 30 °C. In the case of co-inoculation experiments, *S. melonis* Fr1::mTurquoise2 and *P. syringae* DC3000::mScarlet-I were selected on R2A plates containing tetracycline (15 µg mL^−1^) or gentamicin (20 µg mL^−1^), respectively.

### 2.5. Gene Expression Analysis

Six-weeks-old plants were spray-inoculated with either *P. syringae* DC3000, *S. melonis* Fr1 or PBS (mock control), and harvested over time (3 h, 9 h, 48 h). Four mature leaves per plant were collected in an RNase-free 1.5 mL tube and then immediately flash frozen in liquid N_2_. Six plants were sampled per treatment and time point. Flash-frozen samples were ground in the collection tube using Teflon pestles. RNA extraction was performed using the Isolate II RNA Plant kit (Bioline, London, England). During RNA extraction two samples were pooled by using the same lysis buffer on two samples to make up one biological replicate. Purity and quantity of the RNA sample was determined using the Nanodrop (ND-1000 Spectrophotometer, Thermo Scientific, Waltham, MA, USA). Integrity of the RNA sample was assessed via gel electrophoresis. Approximately 1 µg of RNA was used for cDNA synthesis and for the no Reverse Transcriptase (noRT) control, using the VitaScript First strand cDNA synthesis kit (Procomcure Biotech, Thalgau, Austria). RT-qPCR was performed using the 2x ProPlant SYBR Mix (Procomcure Biotech, Thalgau, Austria) in 15 µL reaction volumes with 0.2 µM of each primer and 0.001 g L^−1^ of initial RNA in the cDNA mix. QPCRs were run using the recommended protocol for 2x ProPlant SYBR Mix (Procomcure Biotech, Thalgau, Austria) on a Rotor-Gene Q (Qiagen, Hilden, Germany). Technical triplicates were performed for each biological replicate. The ROX dye, present in the 2x ProPlant SYBR Mix (Procomcure Biotech, Thalgau, Austria), was used to normalise for master mix variation between tubes. A mix of equal amounts of all cDNAs was used for normalisation between runs. mRNA concentrations were calculated using Equation (1).
(1)mRNA concentration a.u.=1primer eff.Cq

Primers that were first used in this study were designed using either the “Universal ProbeLibrary Assay Design Center” (Roche, Basel, Switzerland) or “primer-blast” (NCBI, Bethesda, MD, USA). Primer efficiencies were determined via serial template dilutions [56]. The mRNA concentration of each target gene was then normalised against the mean mRNA concentration of five stably expressed, previously described reference genes (Table 2, [57,58]). Next, the normalised mRNA concentration of each treatment (*P. syringae* DC3000 and *S. melonis* Fr1 inoculation) was normalised against the mean normalised mRNA concentration of mock treated samples to emphasise treatment-related changes in gene expression.

### 2.6. Microscopy and Image Processing

Bacteria were recovered 14 days post-inoculation from the abaxial side of co-inoculated Arabidopsis leaves using the cuticle tape lift procedure described previously [59]. The procedure involves placing one side of the leaf onto double-sided adhesive tape and carefully stripping them off again. Cuticle tape lifts allow the recovery of phyllosphere bacteria from leaves without redistributing their spatial location.

Microscopy was performed using the Zeiss AxioImager.M1 fluorescent widefield microscope (Zeiss, Oberkochen, Germany) at 100x magnification (EC Plan-Neofluar 100x/1.30 PH3 Oil M27 objective) equipped with Zeiss filter sets 38HE (BP 470/40-FT 495-BP 525/50) and 43HE (BP 550/25-FT 570-BP 605/70) for the detection of *S. melonis* Fr1::mTurquoise2 and *P. syringae* DC3000::mScarlet-I, respectively. Acquisition of 3D images was achieved using an Axiocam 506 and the software Zeiss Zen 2.3.

FIJI/ImageJ (Version 2.0.0-rc-54/1.51h) was used for image processing [60]. To improve the signal-to-noise ratio and the depth of field, background subtraction was used before images were z-stacked using maximum intensity projection. The contrast of each resulting image was enhanced.

## 3. Results and Discussion

As shown in Figure 1, Arabidopsis plants were grown in tissue-culture boxes on either agar or zeolite, to compare the suitability of the substrates for gnotobiotic plant growth. Both substrates were covered by a PDMS sheet to reduce the cross-inoculation of phyllosphere and rhizosphere bacteria (Figure 1A,B). Plant growth in the Litterbox system was optimised beforehand to determine optimal amounts of zeolite per tissue-culture box, optimal zeolite granularity, and optimal buffering of growth media (Appendix A). During the three weeks of growth in tissue-culture boxes, the leaves of plants grown on zeolite and agar appeared healthy and dark green (Figure 1C,D). Four-weeks-old plants grown on zeolite were slightly smaller (Figure 1C,D), had significantly fewer leaves (Figure 1L), and weighed significantly less than plants grown on agar (Figure 1K). Noticeably, there was no significant difference in the above-ground organ growth rate of plants grown on agar or zeolite from one week after transfer to tissue-culture boxes (two weeks after sowing) until harvest (Figure 1L; linear regression, *p* = 0.343, ANCOVA). Therefore, the difference in plant weight and the developmental stage was likely caused by a short halt in growth after seedling transfer to zeolite as the plants adapted to a new environment. More importantly, plants grown on zeolite showed a lower degree of variation than plants grown on agar (coefficient of variation: zeolite = 22.77%, agar = 32.92%). This makes the Litterbox system advantageous for predictable and low variability growth during experiments which should result in fewer false positive and negative measurements (Figure 1K).

The most prominent differences in the growth of Arabidopsis on agar compared with zeolite were observed in the rootzone. Tight root networks were observed on the bottoms (Figure 1E) and the sides (Figure 1I) of the Litterbox. In contrast, in the agar system only a few roots penetrated into the agar and those that penetrated did not grow to the bottom of the tissue-culture box (Figure 1F,J). Most of the roots grew on top of the agar (Figure 1H) causing a non-uniform delivery of nutrients to the root surface [36]. Such a plant root phenotype did not occur in the Litterbox system (Figure 1G) which, instead, exhibited root growth similar to soil-grown plants. Furthermore, roots on the agar were noticeably curled (Figure 1H), which is likely linked to a continuous root growth reorientation by the environment-sensing root cap [61]. Predominant growth on top of the agar might be linked to a humidity gradient with the highest water potential on top of the agar, potentially caused by the PDMS sheet [62]. In general, humidity can be adjusted in the Litterbox system by changing the ratio of media to zeolite, whereas in the agar system humidity cannot be adjusted without the agar drying out. Taken together, root growth in the Litterbox system more closely resembled that of a soil-grown plant than that in the agar-based system [63].

Plants that were pre-grown on cut, agar-filled pipette tips (Figure 2A) prior to transfer to zeolite exhibited a lower degree of variation than plants of seeds that were directly sown on zeolite and plants that were transferred without pipette tips (Figure 2B, Coefficient of variation: sown = 34.7%, transferred seedling = 42.28%, transferred seedling on pipette tip = 22.77%). Almost all of the seedlings that were pre-grown in, and then transferred with, the pipette tip survived the transfer to zeolite, whereas only 61.1% of the seedlings that were transferred from an agar plate without pipette tips survived (Figure 2C). Only 84.7% of seeds sown directly on zeolite germinated. Therefore, it is desirable to sow and pre-grow the seeds and seedlings in agar-filled pipette tips, especially when working with transgenic or mutant plant lines that exhibit low germination rates. Different sized pipette tips filled with varying agar strength were tested. Plants pre-grown on pipette tips filled with 1% agar were significantly bigger than plants grown on pipette tips with 0.6% agar strength (Appendix A). Further, plants pre-grown in 200 µL pipette tips exhibited poor survival rates at 0.6% agar (62.5% survived plants), but not at 1% agar (95.83% survived plants) after transfer to zeolite (Appendix A).

Leaves exhibit an environmentally specific carrying capacity for bacterial colonisers [64]. In agar-based systems the bacterial carrying capacity in the phyllosphere is usually around 10^8^ cfu/g [34,35,65]. This is two to four magnitudes greater than the bacterial densities (10^4^–10^6^ cfu/g) of epiphytes found on leaves of various plant species, such as *Arabidopsis thaliana* Col-0 (5 × 10^3–3.5^ × 10^4^ cfu/cm2 ≈ 10^5^–10^6^ cfu/g [66,67]), romaine lettuce, head lettuce, spinach, basil, marjoram, and thyme, grown under temperate environmental conditions [68,69,70]. While the growth season had no effect on the bacterial densities found on leaves of thyme (10^4^ cfu/g) and romaine lettuce (10^5^–10^6^ cfu/g), it did have an effect on the bacterial densities on leaves of basil and marjoram, which started above 10^6^ cfu/g and dropped by almost two magnitudes over the growing period [69,70]. In the Litterbox system, *S. melonis* Fr1, a known biocontrol agent against the model Arabidopsis pathogen *P. syringae* DC3000 reached densities of 10^6^ cfu/g leaf (fresh weight), as opposed to 10^7^ cfu/g leaf when grown on agar (Figure 3). Therefore, the bacterial carrying capacity of leaves in the Litterbox is more comparable to the carrying capacity for epiphytes of leaves of various different plant species grown in temperate environments (10^4^–10^6^ cfu/g) than those observed in agar-based systems.

Besides being designed and geared towards phyllosphere studies, the Litterbox system can be used for a variety of different study designs. In future, it may be of use for rhizosphere studies since bacteria reproduced in the rhizosphere and the zeolite (Figure 4). However, other systems, such as the recently published FlowPot system, which allows rapid changes in media composition, might be advantageous for studies focussing on the rhizosphere [71]. For phyllosphere studies, the Litterbox system is the optimal choice. This is not only because Litterbox-grown Arabidopsis plants compared with agar-grown plants better resemble plants grown in natural environments with regard to phenotype and bacterial densities found on leaves, but also because the use of a PDMS sheet as a physical barrier reduces cross-inoculation between the phyllosphere and rhizosphere. The PDMS sheet significantly reduces the bacterial load in the rhizosphere and the zeolite after spray-inoculation of the phyllosphere by an order of magnitude (Figure 4), while not affecting plant growth (Appendix A). This is especially advantageous for studies that aim to disentangle local from systemic plant responses to microbial colonisation. Plant growth promoting rhizobacteria, for example, were shown to confer broad-spectrum resistance to the entire plant upon rhizosphere colonisation by a process known as induced systemic resistance (ISR) [72,73]. As the PDMS sheet only lowers the bacterial load, but does not prevent microbial colonisation of the rhizosphere, potential systemic responses cannot be fully excluded. However, they are unlikely since even plant growth promoting rhizobacteria need to be present in high densities, above 10^5^ bacteria per gram root (fresh weight), to induce ISR [74].

The Litterbox system allowed us to use a variety of molecular tools to investigate the intimate crosstalk between microbiota members and their host. To determine activation and shaping of the plant immune network, the RNA of the plant host was extracted, and the activation of plant immunity-associated hormones, salicylic acid (SA), ethylene (ET) and jasmonic acid (JA), were estimated using transcript markers previously shown to reflect their respective hormone levels (Figure 5A–C) [59]. As the PDMS sheet lowers the microbial load in the rhizosphere, it is highly likely that changes in transcript-levels are solely attributable to leaf-colonisation. The presence of *P. syringae* DC3000 led to a stronger activation of the ethylene marker compared to *S. melonis* Fr1, even though *P. syringae* DC3000 was present in lower densities, and was previously shown to dampen ethylene production via its HopAF1 effector (Figure 5A,D) [75]. The reduction of the SA marker, pathogenesis-related 1 (PR1), within the first two days of *P. syringae* DC3000 inoculation, as well as the induction of PR1 following *S. melonis* Fr1 inoculation, has been shown previously (Figure 5B) [34]. Transcript levels of the JA marker were downregulated three hours after inoculation with *P. syringae* DC3000 and *S. melonis* Fr1, but upregulated at 48 h after inoculation (Figure 5C), highlighting the need to study plant responses in a time-dependent manner.

The controlled growth conditions for Arabidopsis and associated plant-adapted microbes that the Litterbox system offers also allowed the study of microbe–microbe interactions at the single-cell resolution. For example, we observed the distribution of cell aggregates of *P. syringae* DC3000 and *S. melonis* Fr1 14 days after co-inoculation onto Arabidopsis leaves (Figure 6). Bacterial community members from leaves of Arabidopsis plants that were sampled from a wild population have been shown to establish non-random spatial patterns [59]. As the Litterbox system produces plants that resemble such plants with a comparable bacterial load, it is the ideal system to test the ecological concepts underlying these non-random spatial patterns under laboratory conditions [76]. Bioreporters and single-cell approaches are useful tools to broaden our understanding of bacterial adaptations, colonisation patterns, and physiology in the phyllosphere [1,77,78,79]. Having a growth system that resembles more natural conditions opens the opportunity to study bacterial adaptations to varying environmental factors in a controlled manner (e.g., humidity, light intensity and exposure, and temperature).

## 4. Conclusions

Zeolite can be easily sourced from readily available cat litter making the Litterbox system affordable and easy to establish. The low variation and consistency in plant growth allows for predictable and reproducible experiments, alleviating the chance of false positive and false negative measurements. The Litterbox system is compact, thereby enabling experiments with large sample sizes. Most importantly, plant growth and leaf bacterial densities resemble more closely those found under natural conditions than plants grown in agar-based systems. The Litterbox system enables research on plant–microbe and microbe–microbe interactions and their underlying mechanisms to be conducted in an environmental context. Environmental factors (e.g., humidity, light intensity and exposure, temperature) can be controlled in addition to microbial community composition and plant genotypes. We anticipate that this versatile and affordable gnotobiotic plant growth system will advance microbiota research and broaden our understanding of microbiota assembly, composition, and its effects on the plant host.

## Figures and Tables

**Figure 1 microorganisms-08-00464-f001:**
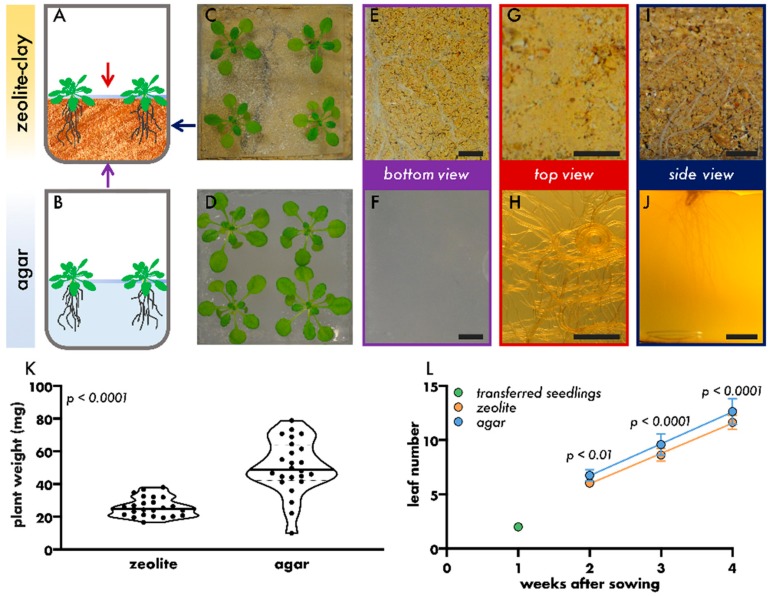
The effect of growth substrate on plant phenotype. (**A**,**B**). Illustration of general growth setup. Blue bar represents a polydimethylsiloxane (PDMS) sheet. Coloured arrows indicate the point of view onto the plant growth box. (**C**,**D**). Representative images of four-weeks-old plants grown on zeolite (**C**) or agar (**D**). (**E**–**J**). Representative images of roots of six-weeks-old plants grown on zeolite (**E**,**G**,**I**) or agar (**F**,**H**,**J**). View on growth box bottom (purple arrow, (**E**,**F**)), top of growth medium (red arrow, (**G**,**H**)) and side of growth box (dark blue arrow, (**I**,**J**)), bar scale = 5 mm. (**K**). Plant weight (fresh weight aboveground plant parts) of four-weeks-old plants grown on zeolite or agar. Dots represent individual samples, thick bar represents median, dotted bars represent quartiles, Mann–Whitney test. (**L**). Number of leaves per plant at different weeks after sowing of plants grown either on zeolite or agar. Seedlings were transferred one week after sowing to either zeolite or agar. Filled circles represent the mean of 24 plants, error bars represent standard deviation, coloured lines represent linear regression lines, Sidak’s multiple comparison test.

**Figure 2 microorganisms-08-00464-f002:**
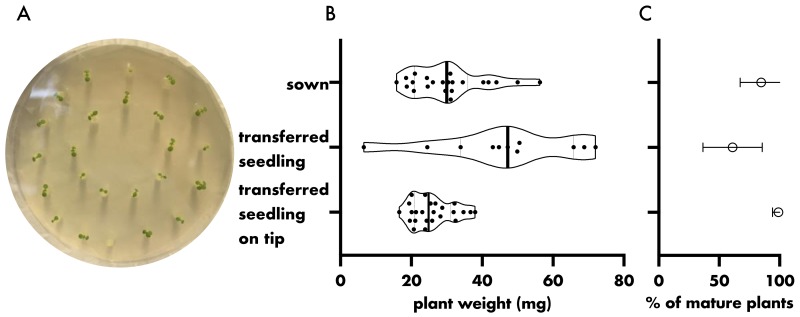
The effect of different sowing strategies on germination rate/seedling survival and plant weight. (**A**). Representative image of seedlings on agar-filled pipette tips seven days after sowing. (**B**,**C**). Plant weight (fresh weight aboveground plant parts) (**B**) and the percentage of mature plants per seed/seedling (**C**) after four weeks of growth. Seeds were either sown directly on zeolite, germinated on agar before transfer to zeolite, or germinated on agar-filled pipette tips before transfer to zeolite. Seedlings were transferred seven days after sowing. Hollow dots represent sample mean, error bars depict standard deviation, filled dots mark individual plants, thick bar represents median, dotted bars represent quartiles.

**Figure 3 microorganisms-08-00464-f003:**
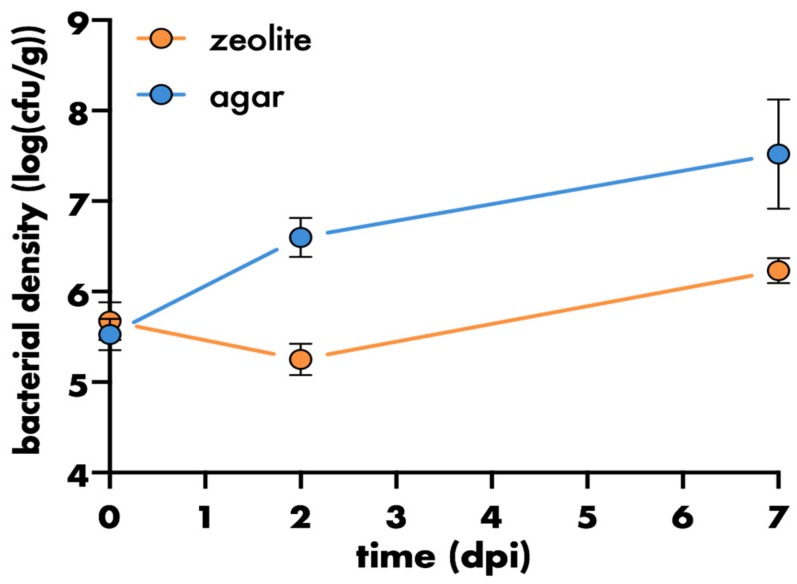
Effect of agar or zeolite clay-based growth substrate on bacterial populations in the phyllosphere. Bacterial density on aboveground plant parts of four-weeks-old plants relative to plant weight (fresh weight of aboveground plant parts). Plants were inoculated with *S. melonis* Fr1. Bacterial populations on agar-grown plants are significantly higher after two and seven days (Sidak’s multiple comparison test, *p* < 0.0001). Filled circles represent the mean of five plants, error bars represent the standard deviation.

**Figure 4 microorganisms-08-00464-f004:**
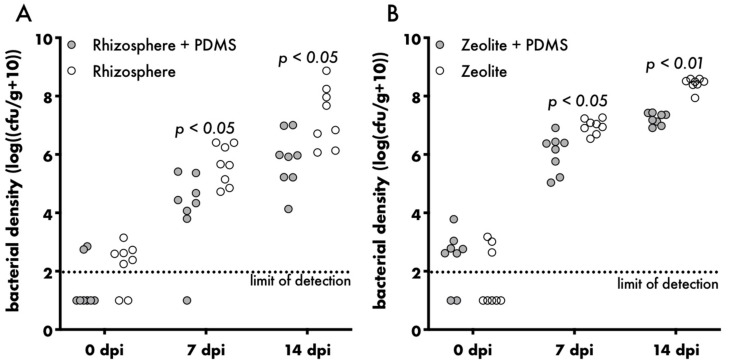
The effect of PDMS on bacterial load in the rhizosphere and zeolite. Bacterial density in the rhizosphere (**A**) or zeolite (**B**) of four-weeks-old plants inoculated with *Sphingomonas melonis* Fr1. Filled circles represent individual samples, dotted line represents the limit of detection, Sidak’s multiple comparison test.

**Figure 5 microorganisms-08-00464-f005:**
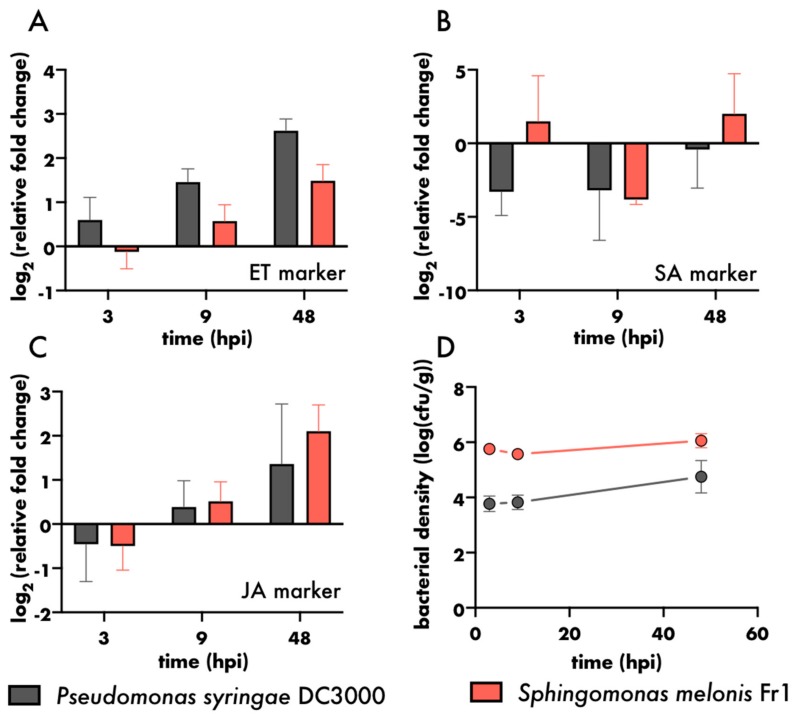
Plant responses to different bacteria. (**A**–**C**). Log_2_ fold change in gene expression relative to mock treated control of (**A**) ethylene marker gene (At2g41230), (**B**) salicylic acid marker gene (PR1) and (**C**) jasmonic acid marker gene (At3g50280). Bars represent the mean of three biological replicates, error bars depict standard deviation. Each biological replicate comprised eight leaves from two plants. (**D**). Bacterial density on aboveground plant parts of six-weeks-old plants relative to plant weight (fresh weight of aboveground plant parts). Filled circles represent mean of six plants, error bars represent standard deviation.

**Figure 6 microorganisms-08-00464-f006:**
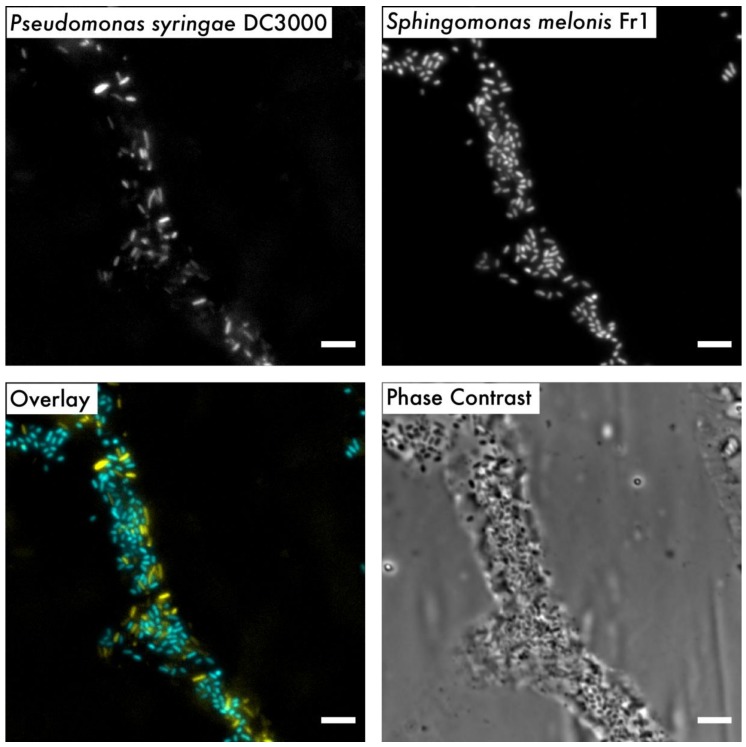
Bacterial distribution on leaves. Representative images of bacterial distribution on leaves after recovery using cuticle tape lifts. Fluorescently tagged *P. syringae* DC3000 and *S. melonis* Fr1 were co-inoculated onto seven-weeks-old Arabidopsis leaves and sampled at 14 days post-inoculation. Overlay image represents the combined pseudo-coloured micrographs of each strain (yellow: *P. syringae* DC3000, cyan: *S. melonis* Fr1). Bar scale 5 µm.

**Table 1 microorganisms-08-00464-t001:** List of bacterial strains used in this study.

Bacterial Strain	Reference	Fluorescence
*Sphingomonas melonis* Fr1	[54]	n.a.
*Pseudomonas syringae* DC3000	[55]	n.a.
*Sphingomonas melonis* Fr1::mTurquoise2	This work	mTurquoise2 (cyan)
*Pseudomonas syringae* DC3000::mScarlet-I	This work	mScarlet-I (red)

**Table 2 microorganisms-08-00464-t002:** List of primers used in this study.

Template	Reference	5′ Primer	3′ Primer
At2g41230 (ET marker)	This study	cgaaccgtccgtacatacataa	ttgcacgaaactaaaactaaaagc
PR1 (SA marker)	This study	gatgtgccaaagtgaggtgtaa	ttcacataattcccacgagga
At3g50280 (JA marker)	This study	ccttcgctggtcgtcttaac	cagagccatcaggtcgaaga
At2g28390 (reference gene)	This study	ggattttcagctactcttcaagcta	tcctgccttgactaagttgaca
At4g26410 (reference gene)	This study	cgtccacaaagctgaatgtg	cgaagtcatggaagccactt
At1g13320 (reference gene)	This study	tgctcagatgagggagagtg	caccagctgaaagtcgctta
EF-1α (reference gene)	This study	cttggtgtcaagcagatgattt	cgtacctagccttggagtatttg
Actin2 (reference gene)	[57]	cttgcaccaagcagcatgaa	ccgatccagacactgtacttcctt

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
