# Peer review of "Litterbox—A gnotobiotic Zeolite-Clay System to Investigate Arabidopsis–Microbe Interactions"

_microorganisms, 2020, doi:10.3390/microorganisms8040464_

Round 1
Reviewer 1 Report
Comments to the Author:
This manuscript described about the application of ‘Litterbox’ system, a highly controlled system to investigate Arabidopsis-microbe interaction. I understand that this manuscript (this study) important in the understanding of mechanisms of interaction between plant and microorganism, and I think that this exploration system may contribute the development of understanding of plant-microbes interaction. However, I think this manuscript has some problem that need correcting.
My specific comments are as follow;
- p3, L110; It is suddenly written by “PDMS sheet”. What is PDMS sheet? Author should describe about the object of using this sheet and the function of this sheet in this section (Introduction).
- p4, L147; Sphingomonas melonis FR1 and Pseudomonas syringae DC3000 were used in this study. Why did the author use these strains? I think that the author needs to show this reason.
- p4, L155; How is wavelength of OD?
- p4, Table 1; Sphingomonas melonis FR1 and Pseudomonas syringae DC3000 which is transfected fluorescence gene describe S. melonis FR1::mTurquoise2 and P. syringae DC3000::mScarlet-I, respectively?
- p4, L178 (2.5. Gene expression analysis) and Figure 5; Are these results of gene expression of Arabidopsis plant using PDMS sheet? I think that the author needs to compare with the result using PDMS sheet and the result unusing PDMS sheet.
- p9, Figure 4; This figure indicates about the result of bacterial density when inoculated S. melonis I think that the author needs to indicate the result of bacterial density when inoculating P. syringae DC3000.
- p10, Figure 5; This figure indicates about the expression of marker gene of ethylene, salicylic acid and jasmonic acid when inoculated S. melonis FR1 and P. syringae DC3000 independently. Why doesn’t the author indicate about the result when co-inoculated these strains?
- p11, Figure 6; I think that the author also needs to indicate about the result of observation of rhizosphere.
Please consider about my suggestions.
Author Response
This manuscript described about the application of ‘Litterbox’ system, a highly controlled system to investigate Arabidopsis-microbe interaction. I understand that this manuscript (this study) important in the understanding of mechanisms of interaction between plant and microorganism, and I think that this exploration system may contribute the development of understanding of plant-microbes interaction. However, I think this manuscript has some problem that need correcting.
We thank the reviewer for their positive feedback regarding the significance of the system to study plant-microbe interactions.
Further, we noticed that the reviewer asked for ‘moderate English changes’. We concluded that some wording might have been convoluted. Therefore, we rephrased some wordings throughout the text to make them better digestible.
My specific comments are as follow;
- p3, L110; It is suddenly written by “PDMS sheet”. What is PDMS sheet? Author should describe about the object of using this sheet and the function of this sheet in this section (Introduction).
We thank the reviewer for pointing out that we need to introduce PDMS already in the introduction and we rephrased the sentence to make it more clear why we use a PDMS sheet in the Litterbox system. The suggestion has been added as requested (L274).
- p4, L147; Sphingomonas melonis FR1 and Pseudomonas syringae DC3000 were used in this study. Why did the author use these strains? I think that the author needs to show this reason.
As the reviewer suggested, we added a rationale for using those two bacterial strains in the discussion (L822-826).
- p4, L155; How is wavelength of OD?
We thank the reviewer for pointing out this mistake. We have added the revised manuscript that we measured OD at 600 nm (OD600nm) (L446).
- p4, Table 1; Sphingomonas melonis FR1 and Pseudomonas syringae DC3000 which is transfected fluorescence gene describe S. melonis FR1::mTurquoise2 and P. syringaeDC3000::mScarlet-I, respectively?
We thank the reviewer for mentioning their issues in understanding table 1. We therefore changed our nomenclature to make it clearer and have added more information about the strains in materials and methods (L451-453).
- p4, L178 (2.5. Gene expression analysis) and Figure 5; Are these results of gene expression of Arabidopsis plant using PDMS sheet? I think that the author needs to compare with the result using PDMS sheet and the result unusing PDMS sheet.
We thank the reviewer for pointing out that it is unclear to them whether the PDMS sheet was used for the gene expression experiment. As the PDMS sheet is an important part of the Litterbox system it was used and we made this clearer throughout the text. A comparison of gene expression changes depending on the presence of a PDMS sheet is in our opinion beyond the scope of this method development paper. However, as mentioned above we discussed the potential induction of systemic responses at lower bacterial loads in more detail in the results/discussion section.
- p9, Figure 4; This figure indicates about the result of bacterial density when inoculated S. melonis I think that the author needs to indicate the result of bacterial density when inoculating P. syringae DC3000.
We humbly disagree with the reviewer. Figure 4 shows the effect of the PDMS sheet in reducing bacterial load in the zeolite and in the rhizosphere. The PDMS sheet works as a physical barrier and, thus, works independent of the applied bacterial strain. We changed the wording in this section to that effect (L853).
- p10, Figure 5; This figure indicates about the expression of marker gene of ethylene, salicylic acid and jasmonic acid when inoculated S. melonis FR1 and P. syringae DC3000 independently. Why doesn’t the author indicate about the result when co-inoculated these strains?
We thank the reviewer for their suggestion. However, we believe that such an experiment is beyond the scope of this study.
- p11, Figure 6; I think that the author also needs to indicate about the result of observation of rhizosphere.
We thank the reviewer for their suggestion. However, we believe that such an experiment is beyond the scope of this study. Further, as mentioned above we changed the text to highlight that the focus of this study is on the phyllosphere.
Please consider about my suggestions.
Thank you very much for your input, your suggestions were all considered
Reviewer 2 Report
The manuscript “Litterbox - A gnotobiotic zeolite-clay system to investigate Arabidopsis-microbe interactions” by Miebach et al. suggested a new soilless medium to study plant microbial interactions using zeolite and a PDMS sheet. Under gnotobiotic conditions, zeolite provided better aeration as compared to agar-based medium, while the PDMS sheet was used to reduce cross-contamination between phyllosphere and rhizosphere. Although other soilless media are available, such as perlite and vermiculite, zeolite could be an alternative material for soilless plant growth media. However, while the authors claim that PDMS sheet can be used as protective layer for plant systemic responses studies, the reduction in cross-contamination is not satisfactory in order to compare local and distal parts of the plant.
Major comments:
Zeolite is apparently a better growth media then agar to study root biology. Please provide root parameters such as root weight and length. Additionally, root-bacteria interactions in agar-based systems are more challenging than leaf-bacteria interactions. Therefore, I strongly advice to demonstrate that the litterbox system can also be used for root interactions. Zeolite has a "nutrient-holding capacity". Are nutrients properly delivered to roots and shoots? The authors claim that plants grown in the litterbox system resemble plants grown in nature (specifically in temperate environments). Please provide more information from literature (plant species, developmental stage). Moreover, an experiment should be included comparing the bacterial load on leaves of Arabidopsis grown on soil, vermiculite/perlite and zeolite. PDMS sheet – there is no significant advantage for this additional material. A reduction of bacterial densities in distal plant parts can still induce local responses. Gene expression/ molecular toods – Is there any difference in gene expression of plants grown on agar or on zeolite?
Minor comment:
4 – rhizosphere means agar?Author Response
Zeolite is apparently a better growth media then agar to study root biology. Please provide root parameters such as root weight and length. Additionally, root-bacteria interactions in agar-based systems are more challenging than leaf-bacteria interactions. Therefore, I strongly advice to demonstrate that the litterbox system can also be used for root interactions.
We thank the reviewer for their positive feedback on zeolite as a growth substitute. Further, we realise that although this study focuses on the phyllosphere (aboveground plant surfaces), the reader’s focus might be drawn too much to the rhizosphere. The system might also be great for rhizosphere studies as mentioned by the reviewer, but as this was not tested in this study, we restructured the text to highlight the phyllosphere and to make it more clear that although the system might be great for rhizosphere studies this was not the scope of the current study and was not tested.
Zeolite has a "nutrient-holding capacity". Are nutrients properly delivered to roots and shoots?
We thank the reviewer for their query regarding nutrient delivery to the plant. In the introduction we mention an uneven nutrient delivery to roots in agar systems, which is linked to root growth on top of the agar. However, we did not readdress this point in the result/discussion section. We agree with the reviewer that this should be pointed out better and included a sentence in the result/discussion section (L715-716).
The authors claim that plants grown in the litterbox system resemble plants grown in nature (specifically in temperate environments). Please provide more information from literature (plant species, developmental stage). Moreover, an experiment should be included comparing the bacterial load on leaves of Arabidopsis grown on soil, vermiculite/perlite and zeolite.
We thank the reviewer for their opinion on the plant phenotype of Litterbox grown plants. To make it more clear we added a few sentences pointing out that the bacterial densities found on different plants differ within two magnitudes and that some change during a growing season, whereas others don’t (L815-838). Further, we highlighted throughout the text that the bacterial load on Litterbox grown plants is still higher, but closer to natural conditions than agar-grown plants.
PDMS sheet – there is no significant advantage for this additional material. A reduction of bacterial densities in distal plant parts can still induce local responses.
We humbly disagree with the reviewer regarding the significance of the PDMS sheet. We show statistically significant magnitude changes in the bacterial load in the zeolite and in the rhizosphere depending on the presence of the PDMS sheet.
We agree with the reviewer, that a lower bacterial load might not prevent the induction of systemic responses in distal plant parts. However, we mention a study showing that plant growth promoting rhizobacteria, that are classic examples for the systemic induction of ISR, need to be present in high densities to induce systemic responses. The presence of this study in key reviews on systemic plant responses shows the general acceptance that high densities are needed (Berendsen et al. 2012; Pieterse et al. 2014). Nevertheless, we agree with the reviewer that systemic responses might still be induced by the lower bacterial load, even though it seems unlikely we therefore further discussed this point in the result/discussion section (L859-862).
Gene expression/ molecular toods – Is there any difference in gene expression of plants grown on agar or on zeolite?
We thank the reviewer for their suggestion. However, we believe that such an experiment is beyond the scope of this study. We show that there are major differences in the phenotype of agar and zeolite grown plants.
The important message of the presented gene expression data is that different bacteria induce different immune responses, and that immune responses are dynamic and should thus be studied in a time-dependent manner.
Round 2
Reviewer 2 Report
The results and discussion section of the manuscript by Miebach et al. has been improved.
However, while the authors included relevant discussion on the bacterial load on the phyllosphere of various plants, they did not mention the bacterial concentration on Arabidopsis leaves based on the literature. Therefore, one of my previous requests remains unaddressed and a simple comparison of the litterbox system with plants grown on a soil-based system was not provided. Additionally, the bacterial load on leaves can simply be adjusted by the initial inoculum in any system selected to grow plants.
Considering that this study shows that the litterbox does provide improvements over agar-based systems [better root development (although it was not the main aim of the study) and reduces bacterial cross-contamination between above- and belowground], I still strongly suggest the authors to fully address the above-mentioned comment.
Author Response
The results and discussion section of the manuscript by Miebach et al. has been improved.
However, while the authors included relevant discussion on the bacterial load on the phyllosphere of various plants, they did not mention the bacterial concentration on Arabidopsis leaves based on the literature.
We concur that a comparison of bacterial densities in the litterbox system to environmentally grown plants is prudent and have added respective studies and comparisons to the manuscript as suggested.
Therefore, one of my previous requests remains unaddressed and a simple comparison of the litterbox system with plants grown on a soil-based system was not provided.
We thank the reviewer for their suggestion and in response, we have exchanged figure 3 for a figure comparing the bacterial load of the model strain Sphingomonas melonis Fr1 recovered from leaves of agar or litterbox-grown plants. Since the previous figure 3 did not add additional important information to the study, we have decided to delete it.
Additionally, the bacterial load on leaves can simply be adjusted by the initial inoculum in any system selected to grow plants.
We humbly disagree with the reviewer. Even though it is true that higher initial inoculum concentration will lead to slightly higher final bacterial loads (Wilson and Lindow 1994), this is only possible within a small range. We have previously shown that leaves do have a carrying capacity for microbes which will lead to bacterial population decrease if too highly concentrated inocula are used (Remus-Emsermann et al. 2012). To clarify this to the reader, we rephrased our wording in line 453 and onwards.
Considering that this study shows that the litterbox does provide improvements over agar-based systems [better root development (although it was not the main aim of the study) and reduces bacterial cross-contamination between above- and belowground], I still strongly suggest the authors to fully address the above-mentioned comment.
We thank the reviewer for their valuable input and their time.
Round 3
Reviewer 2 Report
I have read again manuscript by Miebach et al. (ID: microorganisms-718677) and the revisions made to the manuscript are very effective in addressing the remaining concerns.